# Green Synthesis of Cobalt-Doped CeFe_2_O_5_ Nanocomposites Using Waste *Gossypium arboreum* L. Stalks and Their Application in the Removal of Toxic Water Pollutants

**DOI:** 10.3390/nano14161339

**Published:** 2024-08-12

**Authors:** Saloni Koul, Mamata Singhvi, Beom Soo Kim

**Affiliations:** 1Department of Biotechnology (with Jointly Merged Institute of Bioinformatics and Biotechnology), Savitribai Phule Pune University, Pune 411007, India; n20502022@uopca.unipune.ac.in; 2Department of Chemical Engineering, Chungbuk National University, Cheongju 28644, Chungbuk, Republic of Korea; bskim@chungbuk.ac.kr

**Keywords:** green synthesis, co-precipitation, adsorption, dye removal, pollutant removal, cotton stalks, wastewater treatment, adsorption capacity

## Abstract

Currently, there is an increasing need to find new ways to purify water by eliminating bacterial biofilms, textile dyes, and toxic water pollutants. These contaminants pose significant risks to both human health and the environment. To address this issue, in this study, we have developed an eco-friendly approach that involves synthesizing a cobalt-doped cerium iron oxide (CCIO) nanocomposite (NC) using an aqueous extract of *Gossypium arboreum* L. stalks. The resulting nanoparticles can be used to effectively purify water and tackle the challenges associated with these harmful pollutants. Nanoparticles excel in water pollutant removal by providing a high surface area for efficient adsorption, versatile design for the simultaneous removal of multiple contaminants, catalytic properties for organic pollutant degradation, and magnetic features for easy separation, offering cost-effective and sustainable water treatment solutions. A CCIO nanocomposite was synthesized via a green co-precipitation method utilizing biomolecules and co-enzymes extracted from the aqueous solution of *Gossypium arboreum* L. stalk. This single-step synthesis process was accomplished within a 5-h reaction period. Furthermore, the synthesis of nanocomposites was confirmed by various characterization techniques such as Fourier-transform infrared (FT-IR) spectroscopy, X-ray diffraction (XRD), field emission scanning electron microscopy (FE-SEM), transmission electron microscopy (TEM), thermogravimetric analysis (TGA), dynamic light scattering (DLS), and energy dispersive X-ray (EDX) technology. CCIO NCs were discovered to have a spherical shape and an average size of 40 nm. Based on DLS zeta potential analysis, CCIO NCs were found to be anionic. CCIO NCs also showed significant antimicrobial and antioxidant activity. Overall, considering their physical and chemical properties, the application of CCIO NCs for the adsorption of various dyes (~91%) and water pollutants (chromium = ~60%) has been considered here since they exhibit great adsorption capacity owing to their microporous structure, and represent a step forward in water purification.

## 1. Introduction

Water pollutants pose significant environmental and public health concerns. Environmental consequences include the degradation of aquatic ecosystems, loss of biodiversity, and disruption of ecological balance. Simultaneously, the contamination of water sources increases public health risks, contributing to waterborne diseases, developmental issues, and long-term health problems for communities reliant on a source with compromised water quality. Water streams can be contaminated with a variety of toxic pollutants that include halogenated hydrocarbons, heavy metals, dyes, surfactants, organic compounds, salts, soluble bases, pesticides, and agricultural fertilizers [1]. Synthetic dyes, which are extensively employed in textiles, pharmaceuticals, and numerous other industries, exhibit remarkable color stability and versatility. Hexavalent chromium is a highly toxic heavy metal that is released into water bodies through industrial processes, including metal plating, leather tanning, and textile manufacturing [2]. The ubiquity of these synthetic colorants and metals in water bodies raises alarming concerns as once introduced into aquatic ecosystems, they resist natural degradation processes, leading to their persistence in water bodies, and they can be toxic depending on their chemical composition and concentration [2,3].

Exposure to significant levels of organic/inorganic pollutants has been associated with endocrine disruption, mutagenicity/genotoxicity, and cancer [3,4]. Consequently, nanoparticles emerge as a transformative frontier in the quest for sustainable solutions, holding promise for revolutionizing the landscape of synthetic dye and heavy metal removal [5,6,7,8,9]. In the case of pollutant removal techniques such as flocculation [10,11], ozonation [12], membrane filtration [11,13], activated carbon adsorption [14], electrocoagulation [15,16], sorption techniques [17,18], UV radiation [19], and biological treatment [20], their limitations in terms of efficiency, selectivity, and environmental impact underscore the need for innovative approaches.

Subsequent research focus has shifted to the expanding field of nanoparticle-mediated pollutant removal, addressing the mechanisms by which nanoparticles interact with pollutants and the factors influencing their effectiveness [21]. By reviewing the current state of knowledge, this study aims to underscore the urgency of adopting environmentally benign approaches for pollutant remediation, additionally highlighting the promising role of nanoparticles in mitigating this global environmental challenge [7,22,23]. Au NPs [24], Ag NPs [25,26], TiO_2_ NPs [27], SiO_2_ NPs [28,29], Fe_2_O_3_ NPs [30], and graphene oxide NPs (GO) [31,32] are some of the prime examples of nanomaterial-mediated pollutant remediation technologies.

Nanomaterials, with their increased surface area per unit mass, offer enhanced adsorption capacity. This mechanism involves the physical binding of pollutants to the surface of nanomaterials. The attractive forces between the nanomaterial’s surface and the pollutant molecules lead to their immobilization. This process is particularly effective for the removal of organic pollutants, heavy metals, and even nanoparticles from water. Certain nanomaterials possess catalytic properties, enabling them to facilitate chemical reactions that transform pollutants into less harmful substances. This mechanism is prominent in the degradation of organic pollutants. For instance, photocatalytic nanoparticles, like titanium dioxide (TiO_2_) [27] or zinc oxide (ZnO) [33], can absorb light energy and generate reactive oxygen species, initiating oxidation reactions that break down organic contaminants.

Nanomaterials can induce the precipitation of pollutants by acting as nucleation sites. This is particularly relevant for the removal of heavy metals and ions from water. Functionalized nanoparticles may promote the coagulation of suspended particles, facilitating their removal through sedimentation or filtration processes [34]. Nanomaterials can undergo ion exchange, where certain ions on the material’s surface are replaced by ions from the surrounding water. This mechanism is effective for the removal of specific ions, such as heavy metal ions. Ion exchange properties are often enhanced through surface functionalization, allowing nanomaterials to selectively capture target ions [35,36,37].

Magnetic nanoparticles, which are endowed with magnetic properties, can be separated easily from water using external magnetic fields. This mechanism allows for the recovery and reuse of nanomaterials, contributing to their cost-effectiveness. Magnetic separation is particularly advantageous for the removal of nanoparticles and other magnetic-responsive pollutants. Nanomaterials with redox-active surfaces can participate in redox reactions, facilitating either the reduction or oxidation of pollutants. This mechanism is significant for the removal of contaminants that are susceptible to redox transformations. Redox reactions can be harnessed for the removal of various pollutants, including organic compounds and certain heavy metals [30,38].

Some nanomaterials exhibit inherent antibacterial properties, contributing to the removal of pathogenic microorganisms in water. The disruption of microbial cell membranes or interference with cellular functions by nanomaterials can effectively mitigate waterborne diseases [39,40]. In conclusion, nanomaterials offer a versatile toolkit for pollutant removal in water, employing mechanisms such as adsorption, catalysis, precipitation, ion exchange, magnetic separation, redox reactions, and antibacterial properties. The choice of nanomaterial and the optimization of its properties play a crucial role in designing effective and sustainable water treatment strategies [9]. This study deals with the green synthesis of a cobalt-doped cerium iron oxide (CCIO) nanocomposite (NC) using cotton stalks, a waste material from agriculture that serves as a highly useful, low-cost, and abundantly available material. Green chemistry is a method that aims to reduce waste products and prevent environmental pollution with the aid of sustainable and environmentally friendly materials [8]. It entails the usage of natural resources, such as vegetation, microorganisms, enzymes, and other renewable sources to synthesize various compounds and materials [40,41].

The goal of green synthesis is to develop processes that are secure and green, as well as eco-friendly [26]. In this study, we have utilized cotton stalks as they are agricultural waste; using them as a substrate for nanoparticle deposition aligns with the green chemistry principles of utilizing renewable resources and reducing waste. This approach avoids the need for synthetic or non-renewable materials as substrates. Additionally, the plant extracts are abundant, renewable, and cost-effective compared to synthetic reagents. Hence, we tested this plant waste for the synthesis of CCIO NCs, which proved its effective role in the adsorption of various dyes and toxic metals, in addition to exhibiting potent antimicrobial activity. These properties suggest promising applications for CCIO NCs in the removal of both organic and inorganic pollutants from contaminated water sources. To our knowledge, this is the first study about synthesizing CCIO NCs from *Gossypium arboreum* L. stalks, in terms of their exceptional ability to adsorb contaminants (approximately 91%) via the surface of NCs. The overall work performed in this study can be summarized as shown in Figure 1.

## 2. Materials and Methods

### 2.1. Materials

Cerium nitrate [Ce(NO_3_)_3_], cobalt chloride [CoCl_2_], potassium dichromate [K_2_Cr_2_O_7_], 2,2-diphenyl-1-picrylhydrazyl (DPPH), malachite green (MG), and safranin (SF) were purchased from Himedia, Pune, India. Ferric chloride (FeCl_3_) was obtained from SD Fine Chem, Mumbai, India. Methylene Blue (MB) was bought from SISCO Research Laboratories, Mumbai, India. All other reagents and chemicals were purchased locally.

### 2.2. Synthesis of Cobalt-Doped Cerium Iron Oxide Nanocomposites (CCIO NCs)

The stalks of *Gossypium arboreum* L. were collected from Nagpur, Maharashtra, India. After washing and heat-drying, the stalks were crushed into small pieces. First, 50 g of the crushed stalks were soaked in 500 mL of distilled water and kept in a water bath at 80 °C for 3 h [42]. All the wooden residues were removed by filtration, using a muslin cloth and centrifugation of the extract [8,26,43]. The synthesis of CCIO nanocomposites was performed by co-precipitation, a green synthesis method. In the prepared aqueous extract of *Gossypium arboreum* L. stalks (500 mL), 200 Mm each of Ce(NO_3_)_3_, FeCl_3_, and CoCl_2_ were added stepwise in the mentioned order and kept under stirring at 50 °C for 30 min, then overnight with stirring at 30 °C. A dark-colored solution appeared after overnight stirring and the mixture was further centrifuged at 10,000 rpm for 10 min to obtain the precipitate. The obtained precipitate was further washed in distilled water thrice, followed by ethanol washing. The obtained brown-black-colored pellet was retained for drying at 50 °C.

### 2.3. Characterization of Synthesized CCIO NCs

Fourier-transform infrared (FT-IR Thermo Fisher Scientific, Waltham, MA, USA) spectroscopy was conducted to observe the functional groups, stretching vibrations, and absorption peaks present on the surface of nanocomposites. Field-emission scanning electron microscopy (FE-SEM) analysis was performed to study the morphology of the CCIO NCs. Furthermore, the sizes of the CCIO NCs were estimated using transmission electron microscopy (TEM) (Carl Zeiss, Libra 120 Oberkochen, Germany) at a voltage of 120 Kv [44,45,46]. The elemental composition of the NCs was determined using EDX (Thermo Fisher Scientific, Waltham, MA, USA) analysis. The X-ray diffraction (XRD) [47] pattern of CCIO NCs was obtained using Cu-K_β_ radiation, along with a scintillation counter detector. For smoothening the data, the Savitzky–Golay (SG) digital filtering method was applied. To study the thermostability of the synthesized NCs, thermogravimetric analysis (TGA) was performed. The Malvern Panalytical zeta sizer Nano Z instrument (Malvern, UK) was used to study the zeta potential of the nanocomposites [48].

### 2.4. Preparation of Cationic Dyes

The catalytic properties of CCIO nanocomposites (NCs) for dye decolorization were assessed, as follows. Different concentrations (1, 2, 3, 4, 5, and 10 mg) of CCIO NCs were prepared, to which 10 mL of 100 ppm safranin, malachite green, and methylene blue dyes were added. The solution was stirred for 6 h to check the degradation rate. The dye decolorization process was analyzed by UV–vis spectrophotometer (Thermo Scientific Multiskan EX) [49]. The prepared dye solution (0.1 mg/mL) was used as the control. Eventually, the treatment solution was centrifuged at 7000 rpm for 5 min, and the absorbance (200 nm–800 nm) was measured with a Thermo Fisher Scientific microplate reader. Experiments were performed in triplicate and the mean percentage value was recorded [50,51].

### 2.5. Preparation of Hexavalent Chromium Solution

Analytical grade K_2_Cr_2_O_7_ was dissolved in 10 mL of distilled water to prepare 0.25 mg/mL of chromium sample solution [52]. Based on the standardized concentration of CCIO NCs for dye adsorption, a 4 mg/mL concentration was selected to check its adsorption performance against hexavalent chromium ions.

### 2.6. Adsorption Analysis

To investigate the effect of CCIO NCs on dye as well as for heavy metal removal, the samples were checked for any residual dye/metal at a neutral pH and at room temperature using the absorbance spectra. The adsorption capacity (Q_0_) and percentage of removal (RE) were calculated using the following formulas:
(1)$$Q_{0}=\frac{D_{i}-D_{f}}{W}\times V$$
(2)$$\mathrm{RE}\;\left(\%\right)=\frac{D_{i}-D_{f}}{D_{i}}\times 100$$
where D_i_ and D_f_ are the initial and final absorbance; W is the weight of the adsorbent in g; and V is the volume of the dye solution in liters (L) [53].

### 2.7. Antioxidant Activity

A DPPH assay was employed to measure the scavenging activity of the synthesized NC antioxidants [26,54]. For this assay, 50 mL of 0.1 mM DPPH in methanol was prepared. Then, 0.5 mL of DPPH solution and 0.25 mL of sample solution were mixed and incubated for 30 min at 37 °C. (Additionally, the assay was performed in triplicate, after which the mean value was considered.) After 30 min, absorbance was measured at 517 nm, using the spectrophotometer. Also, the radical scavenging activity was measured using the following equation:(3)$$\mathrm{Radical}\;\mathrm{scavenging}\;\mathrm{activity}\;\left(\%\right)=(\frac{Ac-As}{Ac})\times 100$$
where As = absorbance of the sample, and Ac = absorbance of the control, measured at 517 nm each.

### 2.8. Antimicrobial Activity

The antibacterial activity of CCIO NCs was assessed against *Staphylococcus aureus* and *Escherichia coli* bacteria, using the agar well diffusion method. CCIO NCs were sterilized under UV light for about 30 min. Then 10 mg/mL of suspension was prepared in sterile distilled water and sonicated for 5 min to prepare a homogeneous suspension. Then, 100 µL of bacterial culture was spread over nutrient agar plates. Wells were prepared on both plates using a sterile borer and 50 and 100 µL of CCIO NCs suspension was added to the wells. Then, all the plates were incubated for 24 h at 37 °C. The diameter of the zone of inhibition was then determined to calculate the rate of bacterial growth.

### 2.9. Recycling and Reuse of Used CCIO NCs

The NCs were recovered from the reaction mixture using sonication and Buchner filtration with Whatman filter paper and were then washed with ethanol and double-distilled water, and retained for heat drying [55]. The used NCs were washed 3 times with 70% ethanol. The samples were then dried in a hot air oven for further use. The reusability experiments were carried out using MB dye (5 mg/mL) for 4 succeeding cycles [53].

## 3. Results and Discussion

### 3.1. Synthesis and Characterization of the Synthesized CCIO NCs

During the synthesis of NCs, the plant extract (aqueous), which is added to the metal salt solution (acting as a precursor), facilitates the reduction of metal ions to their zero-valent metallic state. Furthermore, the metal ions nucleate and start to grow into nanoparticles in the presence of stabilizing agents that are present in the plant extract. To understand the structural and morphological characteristics of the synthesized CCIO NCs, SEM and TEM imaging were performed. As shown in Figure 2A, SEM analysis demonstrated that the particles were spherical in shape and were mostly in an aggregated form. The average size of the CCIO NCs was found to be 40 nm using TEM analysis, confirming that the CCIO NCs had been synthesized, as shown in Figure 2B.

Then, EDX analysis was conducted to find out the percentages of elements present in the synthesized CCIO NCs. The position of the peaks from the EDX analysis confirms the elemental composition of the CCIO NCs. As depicted in Figure 2C, the peaks at 4.8 keV, 6.2 keV, and 7.1 keV confirm the presence of cerium, iron, and cobalt elements. In addition, carbon (44.22%) and oxygen (35.48%) were detected (Figure 2C). EDX analysis substantiated the finding that CCIO NCs consist of cerium, cobalt, and iron, with a higher mass percentage of iron than cerium and cobalt [56].

To validate the successful formation of CCIO NCs, FT-IR spectroscopy was administered, as seen in Figure 2D. The FT-IR spectra of CCIO showed an absorption peak at 1373 cm^−1^, which is associated with phenol O–H stretching. The peaks rising at 1235 and 2116 cm^−1^ were attributed to the C–N and C≡C vibrations, respectively. The broad band appearing at 3384.08 cm^−1^ corresponds to the alcohol O–H stretch. The peaks at 1988.46 and 2920.32 cm^−1^ were ascribed to the presence of an aromatic compound and alkane due to the C–H stretch, whereas the peak arising at 1606 cm^−1^ was linked to an alkene (conjugated or cyclic) [43,45,57,58]. The presence of phenols, aromatic compounds, and alkanes in the NCs derived from cotton stalks is a result of the thermal decomposition (via a water bath) of cellulose, lignin, and hemicellulose into smaller molecular fragments, including phenols, aromatic compounds (such as benzene rings), and aliphatic hydrocarbons (alkanes) that are present in the cotton stalks. The zeta potential indicates the stability of the colloidal system and was observed to be −16.8 mV (Figure 2E). The synthesized NCs are anionic and stable, which proves that they have sufficient repulsive force to avoid flocculation [45].

XRD was used to analyze the degree of crystallinity of the synthesized nanocomposites. The samples were scanned in the 2θ range of 15 to 80° and the crystallite size was determined using data obtained during XRD analysis [59]. The XRD diffraction patterns of the CCIO NCs are shown in Figure 2F, and show seven peaks corresponding to the (215), (310), (410), (521), (430), and (300) planes. It is evident that all the peaks showed similarity with the standard pattern, indicating that the synthesized material was pure. The crystallite size was calculated using Scherrer’s formula, which indicated a range of around 35–40 nm, indicating the high crystallinity of CCIO NCs.

TGA was performed to investigate the thermostability of the CCIO NCs. As illustrated in Appendix A, the thermogram showed primary degradation at around 70 °C and, furthermore, degradation at around 200 °C. The weight loss percentage at the second stage of degradation was observed to be higher than the initial one.

### 3.2. Adsorption Studies

#### 3.2.1. Adsorption Performance of CCIO NCs for Cationic Dye Removal

A green co-precipitation method was employed to synthesize CCIO NCs and assess their use for the adsorption of different cationic dyes such as safranin (SF), malachite green (MG), and methylene blue (MB) from aqueous solutions. The CCIO NCs exhibited a maximum adsorption capacity of 34.3 mg g^−1^ when using SF, followed by MB and MG, respectively, thus confirming the possibility of their application in the remediation of dye pollution [60]. The adsorbent was added to the aqueous solution of the dye and kept at room temperature with constant stirring, which allowed the cationic dye to adsorb on the anionic binding site of the CCIO NCs due to electrostatic forces, as seen in Figure 3. Different concentrations of CCIO NCs were added to the aqueous dye solution and UV-vis spectroscopy was performed at the 4th hour. Maximum dye removal was observed when a 5 mg/mL concentration of CCIO NCs was used in the case of all three dyes, as presented in Figure 3A–C. Amongst the dyes used, CCIO NCs exhibited the maximum RE (%) when using SF (92.87%) compared to MB (90.55%) and MG (89.34%), as shown in Figure 3D. Overall, chitosanbased Fe_3_O_4_ NCs were used as adsorbents for MB removal and showed an adsorption capacity of 0.62 to 0.95 mg/g [61]. Li et al. used silica-based chitosan NCs for studying the adsorption behavior of NCs, which demonstrated an adsorption capacity of 43.03 mg g^−1^ for MB [60]. Sadiq et al. synthesized magnetic chitosan deep eutectic solvents (MNCDES) to investigate their adsorption ability regarding MG. These studies showed that MNCDES can adsorb 92.69% of MG dye [60]. In this study, the CCIO NCs showed an almost similar adsorption performance, as reported in previous studies. However, most of the studies have been conducted using chemically synthesized nanomaterials, whereas, in this study, the CCIO NCs have been prepared using plant waste extract through a green synthesis process. Hence, green-synthesized NCs have significant application potential in the removal of dyes from wastewater sources.

#### 3.2.2. Adsorption Performance of CCIO NCs for Chromium Removal

Hexavalent chromium is a tremendously toxic metal of great concern that has been found in water bodies. Several kinds of nanomaterials have been tested as adsorbents for the removal of chromium (VI) from polluted water. However, traditional adsorbents usually have a limited adsorption capacity, which limits their use in real-world applications. In this study, CCIO NCs have been synthesized and studied for chromium (VI) removal from wastewater, owing to their functionality, stability, and redox properties. As shown in Figure 4, the absorbance peak for hexavalent chromium (VI) was observed at 350 nm. The sharpness of this peak decreased significantly when the sample solution was incubated for 6 h at room temperature in the presence of CCIO NCs. CCIO NCs unveiled excellent adsorption capacity for chromium (VI) removal (~59.60%) from a solution, as displayed in Figure 4. From the FTIR analysis, it can be seen that the adsorption effect was due to electrostatic interactions between the surface of the CCIO NCs and chromium (VI). A reduction from chromium (VI) to (III) was ascribed to all the functional groups (e.g., –OH, –COOH, and –NH–) that were present on the CCIO NCs. These results corroborate the conclusion that CCIO NCs have tremendous potential as an economical and efficient adsorbent of chromium (VI) from polluted water.

#### 3.2.3. Comparative Analysis

On the basis of the obtained results, maximum adsorption was obtained until 4 h into the incubation period. Comparative analysis was performed for all three dyes, i.e., SF, MB, and MG, using 10 mg/mL of CCIO concentration. The RE percentage of dye adsorption onto the adsorbent CCIO NCs in 4 h followed the order of SF > MB > MG, as seen in Figure 5. The feasibility of using CCIO NCs in wastewater treatment was investigated by preparing a reaction mixture of dye (10 mg/mL) and the adsorbent. For the adsorption capacity test, the sample was centrifuged and the supernatant was collected to determine the residual dye concentration. Taking into consideration the concentration of CCIO NCs that were utilized for the experiment, the results are highly promising when compared to other studies [52,53,62].

### 3.3. Radical Scavenging Activity

The odd electron of the nitrogen atom in DPPH is reduced by the hydrogen atom that is received from the antioxidants, which reduces the color intensity (violet) (Figure 6). The degree of color reduction was measured spectrophotometrically. As shown in Figure 6, among the different concentrations of NCs, higher antioxidant activity (44%) was observed with a 1.0 mg/mL concentration of CCIO NCs.

### 3.4. Antimicrobial Activity

As shown in Figure 7, the agar diffusion method was used to determine the antimicrobial activity of CCIO NCs. As shown in Table 1, the antimicrobial activity of CCIO NCs was measured against *E. coli* and *S. aureus*. The antimicrobial activity of CCIO NCs was found to be slightly higher in *S. aureus* (19 mm) compared to *E. coli* (16.5 mm). Once a clear zone had formed around the well containing NCs on the media plate, this exhibited proof of inhibition against a particular bacterium.

CCIO NCs. As per the reported studies to date, in addition to well-known antimicrobial AgNPs, other metal oxides (e.g., MgO, TiO_2_, Cu_2_O, ZnO, and CoFe_2_O_4_) have been found to be effective against microorganisms [63,64]. The doping of CeFe_2_O_5_ NPs with cobalt aided in improving their antimicrobial activity against both Gram-positive and Gram-negative organisms, which is evident from previous studies [65]. The excellent antimicrobial activity of CCIO NCs against *E. coli* and *S. aureus* microbes was attributed to the size, functional groups, and charge that were present on the NCs’ surface [66]. The obtained results obviously validate the application of a small amount of CCIO NCs in wastewater treatment, which can reduce the numbers of various types of organisms within a short period of time.

### 3.5. Reusability of CCIO NCs

In addition to good adsorption performance, reusability is a crucial parameter to ensure an efficient, scalable, and economical system. To regenerate the adsorbent, ethanol and methanol were found to be good eluents. Ethanol’s effectiveness in removing dyes such as safranin, methylene blue, and malachite green from nanoparticles is attributed to its ability to disrupt the electrostatic interactions between the dyes and the nanoparticle surfaces. By solvating the dye molecules, ethanol facilitates their removal, enabling the recovery and reuse of the nanoparticles. The effect of four consecutive regeneration cycles was investigated for reused CCIO NCs using SF dye, as described in Section 2.9. The reusability data are expressed with respect to regeneration efficiency (RE). As shown in Figure 8, the CCIO NCs retained almost 85–90% of their adsorption capacity, even after their fourth iteration of recycling. After each successive cycle, barely a 3.5–4% decrease in adsorption performance was observed, indicating that the obtained CCIO NCs had excellent stability, regenerability, and adsorptive properties. Therefore, CCIO can easily be recycled and can act as an excellent cost-effective material for the rapid removal of cationic dyes from an aqueous solution.

### 3.6. Plausible Mechanism of Dye Removal Due to the Action of CCIO NCs

The possible action of CCIO NCs in the removal of dyes and other water pollutants is illustrated in Figure 9. The interaction between dye molecules, such as safranin, methylene blue, malachite green, and CCIO nanocomposites is primarily driven by a combination of electrostatic attraction, hydrogen bonding, and van der Waals forces. The cationic nature of these dyes, due to their dimethylamino groups, facilitates strong electrostatic interactions with negatively charged sites on the nanocomposite’s surface. Additionally, hydrogen bonding plays a crucial role, particularly when the nanocomposite surface contains hydroxyl groups or other hydrogen bond donors or acceptors. The lone pairs on the nitrogen atoms in the dimethylamino groups of the dyes can form hydrogen bonds with the hydrogen atoms of hydroxyl groups on the nanocomposite. Furthermore, the high surface area and porous structure of nanocomposites provide numerous active sites for adsorption, enhancing the van der Waals interactions and allowing for the efficient capture of dye molecules. These combined interactions result in the effective adsorption and removal of dye pollutants from aqueous solutions, making such nanocomposites highly effective in environmental cleanup applications.

## 4. Conclusions

In this study, the green synthesis of CCIO NCs was performed via the co-precipitation method after a short reaction time of 4h. As the synthesized CCIO NCs are free of secondary pollution, cost-effective, and easier to scale up, they may prove to be a better alternative to conventional wastewater treatment techniques. The characteristic features of NCs were studied using various methods, including FT-IR, XRD, TGA, FE-SEM, TEM, and DLS. The results obtained from the FT-IR spectra revealed the presence of phenolic compounds in the CCIO NCs, which established antioxidant activity. The dynamic light scattering (DLS) graph for zeta potential confirms the CCIO NCs to be anionic, which justifies their ability to adsorb cationic dyes like SF, MB, and MG. The TGA curve demonstrates that the synthesized CCIO NPs offer dry heat resistance of up to ± 200 °C. The maximum adsorption capacity of the CCIO NCs was found to be 34.3 mg/g against SF dye. Also, the reused NCs retained the maximum level of 85–90% of their adsorption capacity, even after 4 successive recycling iterations, indicating its reusability. An adsorption capability of CCIO NCs against cationic dyes such as SF, MG, and MB, along with the removal of hexavalent chromium, was determined by using UV-Vis spectroscopy. The results demonstrated a ~91% adsorption of various dyes and ~60% of water pollutants. Hence, considering the environmental factors, CCIO NCs can provide a cost-effective solution against multiple pollutants.

## Figures and Tables

**Figure 1 nanomaterials-14-01339-f001:**
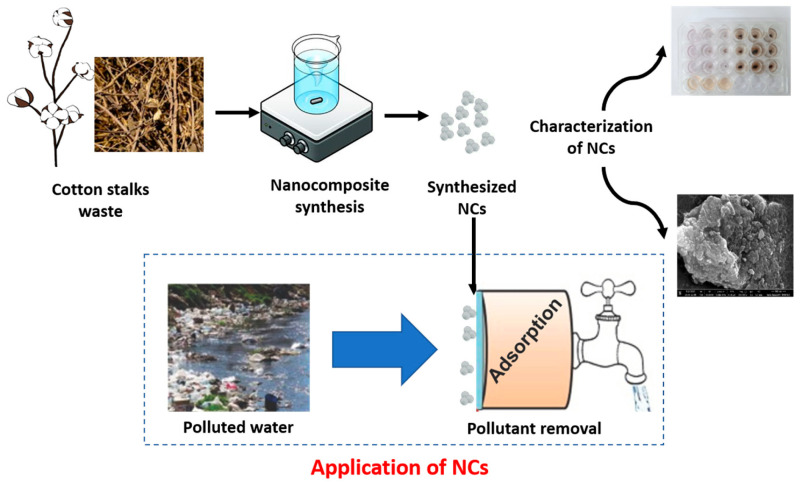
Illustration of the application of synthesized CCIO NCs in water pollutant removal.

**Figure 2 nanomaterials-14-01339-f002:**
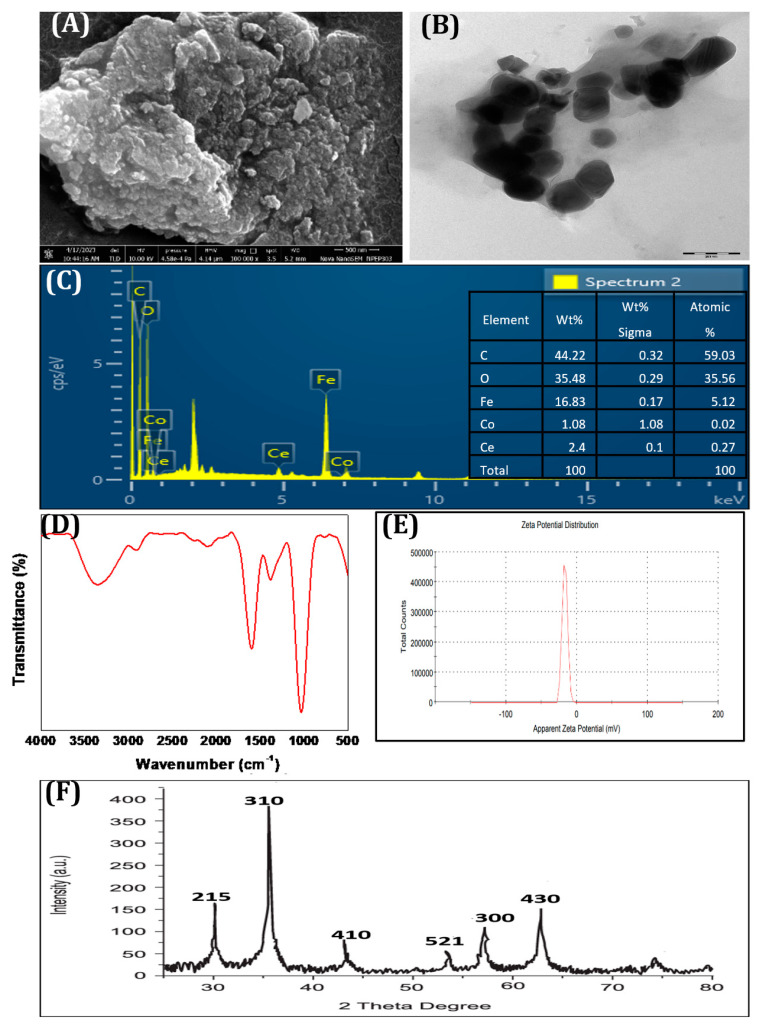
Characterization of CCIO NCs using (**A**) SEM, (**B**) TEM, (**C**) EDX, (**D**) FTIR, (**E**) zeta potential, and (**F**) XRD.

**Figure 3 nanomaterials-14-01339-f003:**
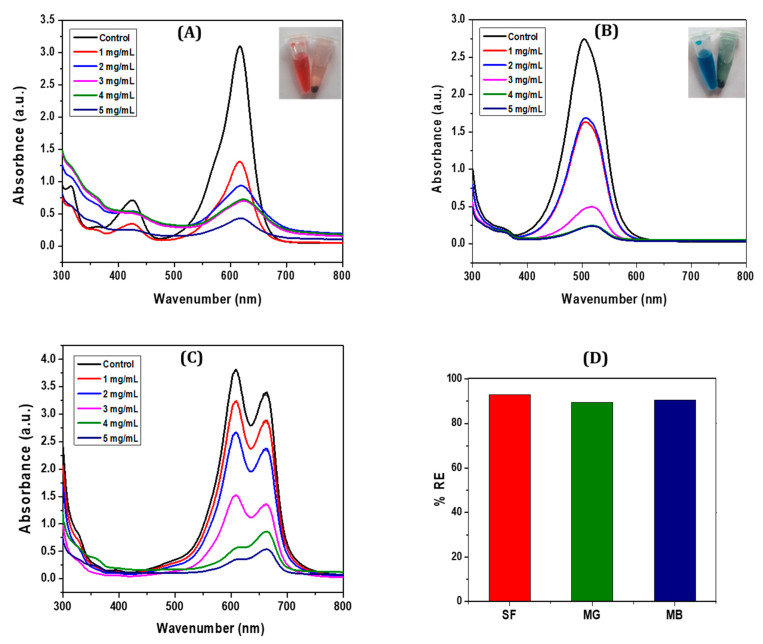
UV-Vis spectra of: (**A**) SF at different concentrations (1–5 mg/mL), (**B**) MG at different concentrations (1–5 mg/mL), (**C**) MB at (1–5 mg/mL), and (**D**) the regeneration efficiency (%) of CCIO NCs using SF, MG, and MB dyes.

**Figure 4 nanomaterials-14-01339-f004:**
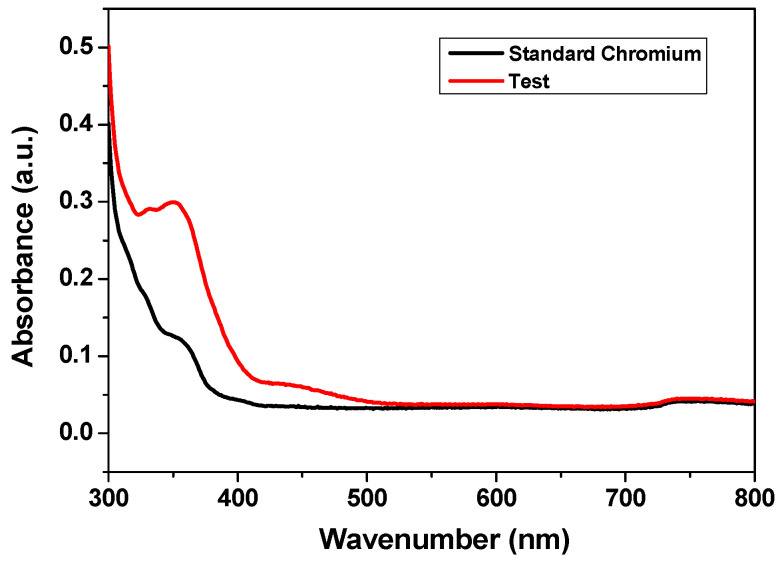
UV-Vis spectra for analyzing the adsorption capacity of CCIO NPs against chromium (VI).

**Figure 5 nanomaterials-14-01339-f005:**
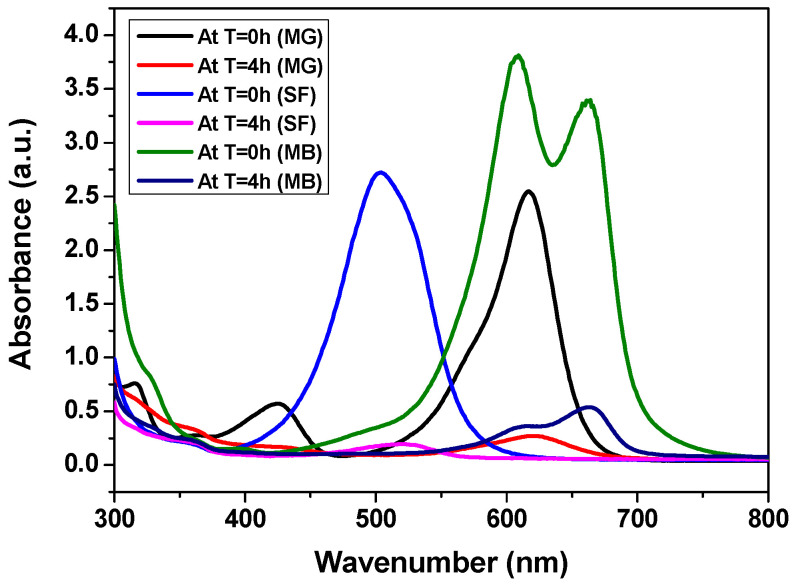
UV-Vis spectra of SF, MG, and MB at 10 mg/mL concentration.

**Figure 6 nanomaterials-14-01339-f006:**
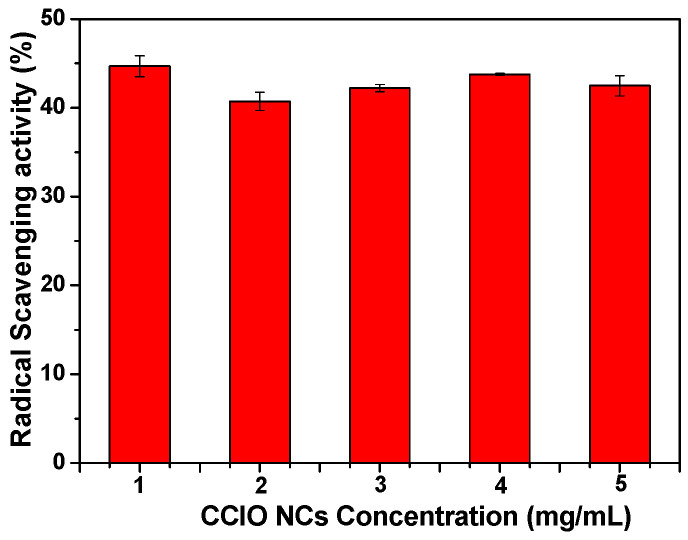
Antioxidant activity of CCIO NCs at different concentrations (1–5 mg/mL).

**Figure 7 nanomaterials-14-01339-f007:**
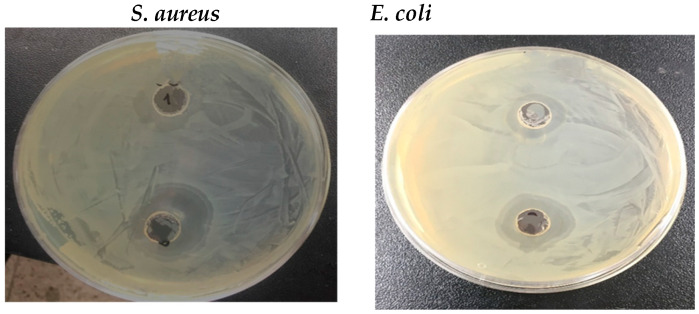
Antimicrobial activity of synthesized CCIO NPs against *E. coli* and *S. aureus*.

**Figure 8 nanomaterials-14-01339-f008:**
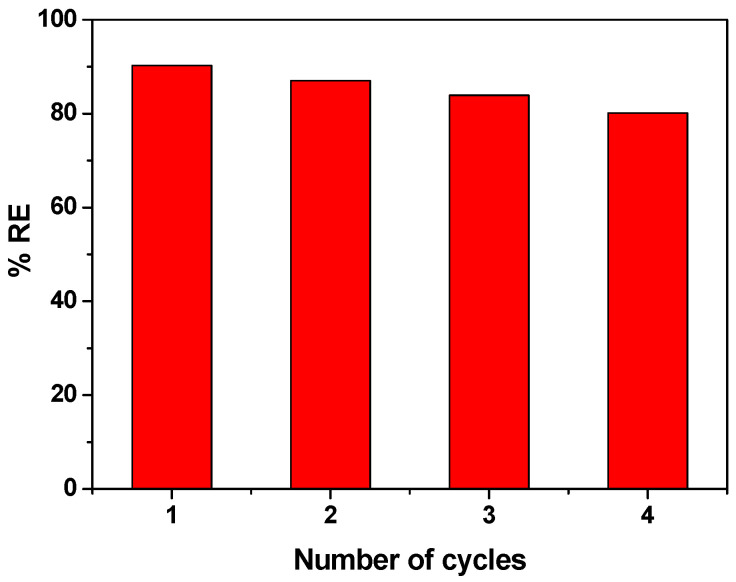
Regeneration efficiency of reused CCIO NCs, using SF dye.

**Figure 9 nanomaterials-14-01339-f009:**
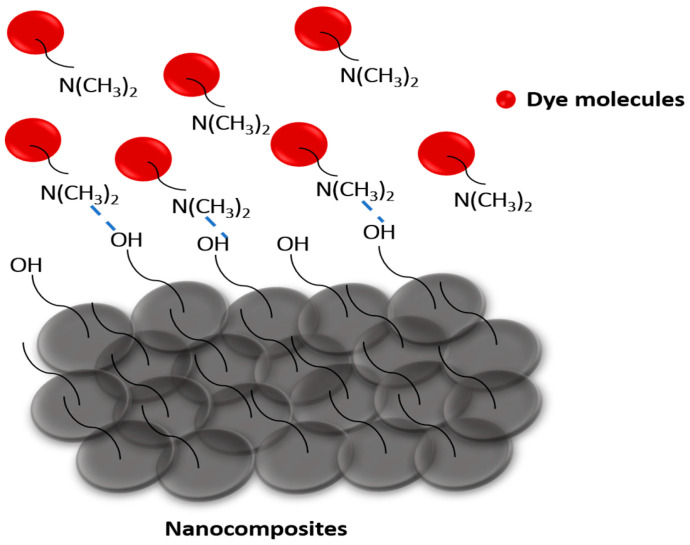
Plausible mechanism of action of CCIO NCs in dye adsorption.

**Table 1 nanomaterials-14-01339-t001:** Antimicrobial effect of CCIO NCs, using the agar well diffusion method.

Microorganisms/Sample	Antimicrobial Activity (Zone of Inhibition (dia. in mm))	
	50 µL	100 µL
*E. coli*	11	16.5
*S. aureus*	13.5	19

## Data Availability

The raw data supporting the conclusions of this article will be made available by the authors upon request.

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
