# Peer review of "Green Synthesis of Cobalt-Doped CeFe_2_O_5_ Nanocomposites Using Waste *Gossypium arboreum* L. Stalks and Their Application in the Removal of Toxic Water Pollutants"

_nanomaterials, 2024, doi:10.3390/nano14161339_

Round 1
Reviewer 1 Report
Comments and Suggestions for Authors
The paper reports on a new material for water purification acting antimicrobial and as antioxidant. The Cobalt-doped Cerium Iron Oxide is deposited on from cotton stalk trying to establish a green route for a water purification material. But is this really a green route, if all active inorganic materials have to be deposited on the cotton stalk – I very much doubt this.
Moreover, the paper lacks on some important aspects, which are currently neglected. To improve their paper, the authors must consider the following points in a revised version:
1. Chapter 2.2 – It must be clarified what the advantages of the use of “cotton stalk” are. Why is this a favored material; what other waste biomasses could be used as well?
What are the ecological and economic balances for the utilization of different waste biomasses?
2. Chapters 2.7 and 2.8 – It is necessary to give more detailed information about the sensitivity of the DPPH assay. Furthermore, some literature giving information about the sensitivity levels of Staphylococcus aureus and Escherichia coli bacteria should be cited as well.
3. Chapter 3.1: Before giving results of the formed CCIO NCs I would expect an analysis of the cotton stalks. I expect that inorganic species are also present in that (most plants contain inorganics as well); looking from them requires a careful analysis of the ash of cotton stalks. This analysis is important since the inorganic components of the cotton stalk might interfere with the inorganic compounds added in the synthesis off the CCIO-NCs.
4. Chapter 3.1: in line 139 it is only said, that the “stalks were crushed in small pieces”. Here now 40 nm CCIO NCs were found. Is there any relation between the extent of crushing of the stalks are the size of the formed NCs.
With other words, how is the influence of the stalks on the NCs, i.e. what is the mechanism defining the NC size?
5. Chapter 3.2.1 and 3.5: In lines 316/317 it is said that the dyes adsorb by electrostatic forces. Thus, the adsorption is reversible. In Chapter 3.5 it is said, that the CCIO NCs regenerated with methanol or ethanol. However, I miss information, (i) how much alcohol is needed for the regeneration and (ii) why the alcohol can dissolve the dye from the NCs?
6. Chapter 3.2.1: Chitosan deep eutectic solvents (MNCDES) are somewhat more efficient in binding dyes than CCIO NCs. The authors should argue why, i.e. due to which interactions, this is the case.
Moreover, other “green synthesized” materials might also be possible. It is not obvious cotton stalks might be suited best.
7. Chapter 3.2.2: The time-dependence of the chromium (VI) adsorption remains unclear; more data (for different exposition times) are required in Fig. 4. So that the adsorption kinetics can be determined and can be compared to those of other adsorption materials.
Moreover, it must be tested, how the chromium can be removed from the CCIO NCs to regenerate the sorbent. This is not mentioned in Chapter 3.5, but must be added there.
8. Conclusions and general: The CCIO NCs consist of CeFe2O5 and are Co-doped. In the whole paper I do not learn, why cerium ferrite is needed and what effect the Co-doping has.
This must be clearly explained, since other ferrites might be possible as well and because doping is always another additional step increasing the effort of NC formation.
Comments on the Quality of English Language
Line 139- it must be “50 mg”.
No further distinct comments.
Author Response
Reviewer 1
Q1. The paper reports on a new material for water purification acting antimicrobial and as an antioxidant. The Cobalt-doped Cerium Iron Oxide is deposited on from cotton stalk trying to establish a green route for a water purification material. But is this really a green route, if all active inorganic materials have to be deposited on the cotton stalk – I very much doubt this.
Ans. Thank you so much for your valuable comments. To clarify, we are not aiming at depositing the inorganic materials on the cotton stalk. The metal salts (ferric chloride, cobalt chloride, cerium nitrate) are a precursor solution. The plant extract (aqueous), which is added to the metal salt solution facilitates the reduction of metal ions to their zero-valent metallic state. Once reduced, the metal ions nucleate and start to grow into nanoparticles in the presence of stabilizing agents present in the plant extract.
Q2. Chapter 2.2 – It must be clarified what the advantages of the use of “cotton stalk” are. Why is this a favored material; what other waste biomasses could be used as well?
What are the ecological and economic balances for the utilization of different waste biomasses?
Ans. Thank you for your suggestion. The experiment was conducted in Maharastra, India; being a prominent cotton-producing state in India, which has a significant availability of cotton stalks as agricultural waste. Cotton stalks are generated in large quantities during cotton harvesting seasons, providing ample biomass for various applications including nanomaterial synthesis. Other waste biomasses that could be used including- Rice husk. Sugarcane bagasse, Corn stover etc. The changes have been made, and highlighted in yellow color.
Q3. Chapters 2.7 and 2.8 – It is necessary to give more detailed information about the sensitivity of the DPPH assay. Furthermore, some literature giving information about the sensitivity levels of Staphylococcus aureus and Escherichia coli bacteria should be cited as well.
Ans. We performed a DPPH assay to determine whether the synthesized NCs have antioxidant properties or not. This is an initial part of the characterization process that can aid in identifying the potential applications of NCs. We used different concentrations of NCs and found that the lowest concentration (1.0 mg/mL) exhibited 44% antioxidant activity, which remained almost constant even after increasing the concentration of NCs. We used the same concentration of NCs to test for antimicrobial activity, which showed a zone of clearance, as described in the manuscript. Increasing the concentration of NCs did not lead to an increase in the zone of inhibition.
Q4. Chapter 3.1: Before giving results of the formed CCIO NCs I would expect an analysis of the cotton stalks. I expect that inorganic species are also present in that (most plants contain inorganics as well); looking for them requires a careful analysis of the ash of cotton stalks. This analysis is important since the inorganic components of the cotton stalk might interfere with the inorganic compounds added in the synthesis off the CCIO-NCs.
Ans. Thank you for your valuable suggestion. We have performed the biocompositional analysis of cotton stalk and explained in revised MS in supplementary information. Table S1 displaying biocomposition analysis has been included in the revised mS.
Q5. Chapter 3.1: in line 139 it is only said, that the “stalks were crushed in small pieces”. Here now 40 nm CCIO NCs were found. Is there any relation between the extent of crushing of the stalks are the size of the formed NCs.
With other words, how is the influence of the stalks on the NCs, i.e. what is the mechanism defining the NC size?
Ans. While the size of the chopped plant material used in nanoparticle synthesis is not a direct determinant of nanoparticle size, it is the composition and properties of the plant extract and the synthesis conditions that primarily influence the characteristics of the synthesized nanoparticles. One can optimize these parameters to achieve nanoparticles with desired size and properties for specific applications.
Q6. Chapter 3.2.1 and 3.5: In lines 316/317 it is said that the dyes adsorb by electrostatic forces. Thus, the adsorption is reversible. In Chapter 3.5 it is said, that the CCIO NCs regenerated with methanol or ethanol. However, I miss information, (i) how much alcohol is needed for the regeneration and (ii) why the alcohol can dissolve the dye from the NCs?
Ans. (i) The NCs are to be soaked in the alcohol until there is no residual dye, bleeding into the freshly added ethanol. (ii) Ethanol's polar nature facilitates the dissolution (solvation) of cationic dye molecules like safranin, methylene blue, and malachite green. The solvation process disrupts the electrostatic interactions between the dyes and the NC surface, allowing the dyes to detach from the nanoparticles. Therefore, Ethanol serves as an effective medium to remove and recover the dyes from the nanoparticle-bound complexes. The changes are highlighted in yellow color.
Q7. Chapter 3.2.1: Chitosan deep eutectic solvents (MNCDES) are somewhat more efficient in binding dyes than CCIO NCs. The authors should argue why, i.e. due to which interactions, this is the case.Moreover, other “green synthesized” materials might also be possible. It is not obvious cotton stalks might be best suited.
Ans. Yes, We agree with the Reviewer’s comment about the other sources might be suited best for dye removal. We also tried sugarcane bagasse as one of the waste for synthesizing nanomaterials but we could not see formation of NPs. As we have mentioned above that cotton stalks are complete waste and can not be used for any kind of application. Hence, we thought of utilizing this substrate which showed promising results.
Q8. Chapter 3.2.2: The time-dependence of the chromium (VI) adsorption remains unclear; more data (for different exposition times) are required in Fig. 4. So that the adsorption kinetics can be determined and can be compared to those of other adsorption materials.
Moreover, it must be tested, how the chromium can be removed from the CCIO NCs to regenerate the sorbent. This is not mentioned in Chapter 3.5, but must be added there.
Ans. Thank you for your feedback. Initially, we standardized the conditions for the adsorption of different dyes using CCIO NCs. We then applied the optimized conditions to examine the role of NCs in chromium removal. This was an initial test to assess the effectiveness of NCs in removing water pollutants such as chromium. Our future studies will concentrate on a detailed investigation of process optimization on both small and large scales using different water pollutants.
Q9. Conclusions and general: The CCIO NCs consist of CeFe2O5 and are Co-doped. In the whole paper I do not learn, why cerium ferrite is needed and what effect the Co-doping has.
This must be clearly explained, since other ferrites might be possible as well and because doping is always another additional step increasing the effort of NC formation.
Ans. There are always possibilities to see variations in performance when we use different kinds of nanomaterials. Here, we synthesized cerium and iron-based nanocomposites as we have studied previously about various enzyme mimicking properties of cerium iron oxide NPs. And also several functional groups attached during green synthesis of NCs which might play a role in strong interaction between dyes and NCs. Ultimately the maximum water pollutants or dyes can be removed effectively. Hence, we aimed to apply CCIO NCs in the current study.
Q10. Line 139- it must be “50 mg”.
Ans. Thank you for your comments. There was a mistake between mg and g. The MS has been modified and highlighted in yellow color.

Reviewer 2 Report
Comments and Suggestions for Authors
Comments
In this work, the authors an eco-friendly approach that involves synthesizing Cobalt-doped Cerium Iron Oxide (CCIO) nanocomposite (NC) using an aqueous extract of Gossypium arboreum L. stalks. The resulting nanoparticles can be used to effectively purify water and tackle the challenges associated with harmful pollutants. In my view, the following issues need to be addressed before consideration for publication.
1. Please increase the resolution of all images in the text.
2. Please check the description of the infrared spectrum and explain the origin of phenol, aromatic compound and alkane in your sample.
3. Please provide additional information on the intrinsic mechanism for the good adsorption properties of the prepared samples.
4. It is recommended that you refer to high level quality articles for the plotting and placement of diagrams.
5. Please provide a description of the role of waste Gossypium arboreum L. stalk.
Comments on the Quality of English Language
Minor editing of English language required
Author Response
Reviewer 2
In this work, the authors an eco-friendly approach that involves synthesizing Cobalt-doped Cerium Iron Oxide (CCIO) nanocomposite (NC) using an aqueous extract of Gossypium arboreum L. stalks. The resulting nanoparticles can be used to effectively purify water and tackle the challenges associated with harmful pollutants. In my view, the following issues need to be addressed before consideration for publication.
Q1. Please increase the resolution of all images in the text.
Ans. As per suggested, we have improved resolutions of all images in the revised MS.
Q2. Please check the description of the infrared spectrum and explain the origin of phenol, aromatic compound, and alkane in your sample.
Q3. Please provide additional information on the intrinsic mechanism for the good adsorption properties of the prepared samples.
Ans. Yes, we have now included the complete section describing the mechanism of action of CCIO NCs in dye removal. Section 3.6 and Figure 9 is added in the revised MS
Q4. It is recommended that you refer to high level quality articles for the plotting and placement of diagrams.
Ans. Thank you for your comments. Yes, we have changed the placement of diagrams in the revised MS.
- Please describe the role of waste Gossypium arboreum L. stalk.
Ans. The plant extract (aqueous), which is added to the metal salt solution (acts as a precursor) facilitates the reduction of metal ions to their zero-valent metallic state. Further, the metal ions nucleate and start to grow into nanoparticles in the presence of stabilizing agents present in the plant extract. This information is included in the revised MS (section 3.1) and highlighted in yellow color.

Reviewer 3 Report
Comments and Suggestions for Authors
The current manuscript is evidently in its preliminary stage, prompting the authors to consider the following enhancements:
1. In the context of Cobalt-doped CeFe2O5, it is imperative to discern between Fe2O5 and Fe3O4 phases.
2. While the introduction offers a comprehensive overview, the focus on magnetic composites and toxic metals should be prioritized over nanomaterials alone.
3. Inclusion of XRD patterns within the main manuscript is recommended, accompanied by a thorough discussion and clear labeling of XRD peaks.
4. The addition of both low and high magnification SEM images, TEM images, and STEM analysis is essential to substantiate the nanocomposite properties.
5. What about the size shape modification after doping
6. Consideration of Dynamic Light Scattering study would provide valuable insights into particle size distribution and stability.
7. A more detailed investigation into antimicrobial activity beyond inhibition zones is warranted.
Comments on the Quality of English Language
nil
Author Response
Reviewer 3
The current manuscript is evidently in its preliminary stage, prompting the authors to consider the following enhancements:
Q1.In the context of Cobalt-doped CeFe2O5, it is imperative to discern between Fe2O5 and Fe3O4 phases.
Ans. The linear formula for the synthesized Cobalt-doped CeFe2O5 is as: CeO2:Fe2O3 and the studies done by Ning et al., 2021 warrant the presence of Fe2O3 phases in such NCs.
Q2.While the introduction offers a comprehensive overview, the focus on magnetic composites and toxic metals should be prioritized over nanomaterials alone.
Ans. Thank you for the reviewer’s valuable suggestions. In lines 37-53, we have focused on the toxic pollutants and their repercussions, further providing an insight into different nanomaterials and their working mechanisms, helping the reader understand the mechanism of CCIO NCs that function on a similar principle.
Q3.Inclusion of XRD patterns within the main manuscript is recommended, accompanied by a thorough discussion and clear labeling of XRD peaks.
Ans. As per suggested, XRD figure is included in the main manuscript and labelling of XRD peaks have been done. The inclusions are highlighted in yellow color in revised MS.
Q4. The addition of both low and high-magnification SEM images, TEM images, and STEM analysis is essential to substantiate the nanocomposite properties.
Ans. We have selected the best images and presented in the main MS.
Q5.What about the size shape modification after doping?
Ans. Thank you for comments. As compared to the studied reported by Ning et al., 2021, (who worked on the synthesis of Cerium Iron Oxide NC) cobalt doping doesn’t cause significant changes in the structure but promotes soft agglomeration that increases particle size.
- Consideration of Dynamic Light Scattering study would provide valuable insights into particle size distribution and stability.
Ans. Thank you for your comments, please refer to Figure 2E.
- A more detailed investigation into antimicrobial activity beyond inhibition zones is warranted.
Ans. Our goal was to test the effectiveness of NCs in removing various pollutants, contaminants, and dyes. Therefore, we focused mainly on these aspects. Additionally, we conducted antimicrobial activity tests to assess the inhibition of microbes in polluted or contaminated water. We tested the antimicrobial activity against both Gram-positive and Gram-negative microbes, which provided us with a good understanding of the concentrations that can inhibit microbial growth. In future studies, we are going to conduct studies at larger scale with more optimization studies, that time we may study in detail about antimicrobial activity.
